# Human-in-the-Loop Interpretability Prior

**Isaac Lage**
Department of Computer Science
Harvard University
isaaclage@g.harvard.edu

**Andrew Slavin Ross**
Department of Computer Science
Harvard University
andrew_ross@g.harvard.edu

**Been Kim**
Google Brain
beenkim@google.com

**Samuel J. Gershman**
Department of Psychology
Harvard University
gershman@fas.harvard.edu

**Finale Doshi-Velez**
Department of Computer Science
Harvard University
finale@seas.harvard.edu

## Abstract

We often desire our models to be interpretable as well as accurate. Prior work on optimizing models for interpretability has relied on easy-to-quantify proxies for interpretability, such as sparsity or the number of operations required. In this work, we optimize for interpretability by *directly* including humans in the optimization loop. We develop an algorithm that minimizes the number of user studies to find models that are both predictive and interpretable and demonstrate our approach on several data sets. Our human subjects results show trends towards different proxy notions of interpretability on different datasets, which suggests that different proxies are preferred on different tasks.

## 1   Introduction

Understanding machine learning models can help people discover confounders in their training data, and dangerous associations or new scientific insights learned by their models [3, 9, 15]. This means that we can encourage the models we learn to be safer and more useful to us by effectively incorporating interpretability into our training objectives. But interpretability depends on both the subjective experience of human users and the downstream application, which makes it difficult to incorporate into computational learning methods.

Human-interpretability can be achieved by learning models that are inherently easier to explain or by developing more sophisticated explanation methods; we focus on the first problem. This can be solved with one of two broad approaches. The first *defines* certain classes of models as inherently interpretable. Well known examples include decision trees [9], generalized additive models [3], and decision sets [13]. The second approach identifies some *proxy* that (presumably) makes a model interpretable and then optimizes that proxy. Examples of this second approach include optimizing linear models to be sparse [29], optimizing functions to be monotone [1], or optimizing neural networks to be easily explained by decision trees [33].

In many cases, the optimization of a property can be viewed as placing a prior over models and solving for a MAP solution of the following form:

$$\max_{M \in \mathcal{M}} p(X|M)p(M) \tag{1}$$

where $\mathcal{M}$ is a family of models, $X$ is the data, $p(X|M)$ is the likelihood, and $p(M)$ is a prior on the model that encourages it to share some aspect of our inductive biases. Two examples of biases include the interpretation of the L1 penalty on logistic regression as a Laplace prior on the weights and the

class of norms described in Bach [2] that induce various kinds of structured sparsity. Generally, if we have a functional form for $p(M)$, we can apply a variety of optimization techniques to find the MAP solution. Placing an interpretability bias on a class of models through $p(M)$ allows us to search for interpretable models in more expressive function classes.

Optimizing for interpretability in this way relies heavily on the assumption that we can quantify the subjective notion of human interpretability with some functional form $p(M)$. Specifying this functional form might be quite challenging. In this work, we *directly* estimate the interpretability prior $p(M)$ from human-subject feedback. Optimizing this more direct measure of interpretability can give us models more suited to a task at hand than more accurately optimizing an imperfect proxy.

Since measuring $p(M)$ for each model $M$ has a high cost—requiring a user study—we develop a cost-effective approach that initially identifies models $M$ with high likelihood $p(X|M)$, then uses model-based optimization to identify an approximate MAP solution from that set with few queries to $p(M)$. We find that different proxies for interpretability prefer different models, and that our approach can optimize all of these proxies. Our human subjects results suggest that we can optimize for human-interpretability preferences.

## 2 Related Work

**Learning interpretable models with proxies**   Many approaches to learning interpretable models optimize proxies that can be computed directly from the model. Examples include decision tree depth [9], number of integer regression coefficients [30], amount of overlap between decision rules [13], and different kinds of sparsity penalties in neural networks [10, 24]. In some cases, optimizing a proxy can be viewed as MAP estimation under an interpretability-encouraging prior [29, 2]. These proxy-based approaches assume that it is possible to formulate a notion of interpretability that is a computational property of the model, and that we know a priori what that property is. Lavrac [14] shows a case where doctors prefer longer decision trees over shorter ones, which suggests that these proxies do not fully capture what it means for a model to be interpretable in all contexts. Through our approach, we place an interpretability-encouraging prior on arbitrary classes of models that depends directly on human preferences.

**Learning from human feedback**   Since interpretability is difficult to quantify mathematically, Doshi-Velez and Kim [8] argue that evaluating it well requires a user study. Many works in interpretable machine learning have user studies: some advance the science of interpretability by testing the effect of explanation factors on human performance on interpretability-related tasks [20, 19] while others compare the interpretability of two classes of models through A/B tests [13, 11]. More broadly, there exist many studies about situations in which human preferences are hard to articulate as a computational property and must be learned directly from human data. Examples include kernel learning [28, 31], preference based reinforcement learning [32, 5] and human based genetic algorithms [12]. Our work resembles human computation algorithms [16] applied to user studies for interpretability as we use the user studies to *optimize* for interpretability instead of just comparing a model to a baseline.

**Model-based optimization**   Many techniques have been developed to efficiently characterize functions in few evaluations when each evaluation is expensive. The field of Bayesian experimental design [4] optimizes which experiments to perform according to a notion of which information matters. In some cases, the intent is to characterize the entire function space completely [34, 17], and in other cases, the intent is to find an optimum [27, 26]. We are interested in this second case. Snoek *et al.* [26] optimize the hyperparameters of a neural network in a problem setup similar to ours. For them, evaluating the likelihood is expensive because it requires training a network, while in our case, evaluating the prior is expensive because it requires a user study. We use a similar set of techniques since, in both cases, evaluating the posterior is expensive.

## 3 Framework and Modeling Considerations

Our high-level goal is to find a model $M$ that maximizes $p(M|X) \propto p(X|M)p(M)$ where $p(M)$ is a measure of human interpretability. We assume that computation is relatively inexpensive, and thus computing and optimizing with respect to the likelihood $p(X|M)$ is significantly less expensive

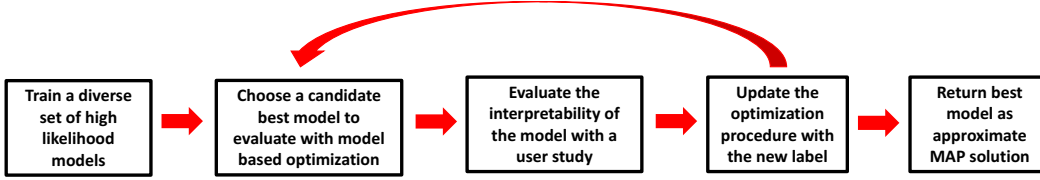

Figure 1: High-level overview of the pipeline

than evaluating the prior $p(M)$, which requires a user study. Our strategy will be to first identify a large, diverse collection of models $M$ with large likelihood $p(X|M)$, that is, models that explain the data well. This task can be completed without user studies. Next, we will search amongst these models to identify those that also have large prior $p(M)$. Specifically, to limit the number of user studies required, we will use a model-based optimization approach [27] to identify which models $M$ to evaluate. Figure 1 depicts the steps in the pipeline. Below, we outline how we define the likelihood $p(X|M)$ and the prior $p(M)$; in Section 4 we define our process for approximate MAP inference.

## 3.1 Likelihood

In many domains, experts desire a model that achieves some performance threshold (and amongst those, may prefer one that is most interpretable). To model this notion of a performance threshold, we use the soft insensitive loss function (SILF)-based likelihood [6, 18]. The likelihood takes the form of

$$p(X|M) = \frac{1}{Z} e^{(-C \times \texttt{SILF}_{\epsilon,\beta}(1 - \texttt{accuracy}(X,M)))}$$

where $\texttt{accuracy(X,M)}$ is the accuracy of model $M$ on data $X$ and $\texttt{SILF}_{\epsilon,\beta}(y)$ is given by

$$\texttt{SILF}_{\epsilon,\beta}(y) = \begin{cases} 0, & 0 \leq y \leq (1-\beta)\epsilon \\ \frac{(y-(1-\beta)\epsilon)^2}{4\beta\epsilon}, & (1-\beta)\epsilon \leq y \leq (1+\beta)\epsilon \\ y - \epsilon, & y \geq (1+\beta)\epsilon \end{cases}$$

which effectively defines a model as having high likelihood if its accuracy is greater than $1 - (1-\beta)\epsilon$.

In practice, we choose the threshold $1 - (1-\beta)\epsilon$ to be equal to an accuracy threshold placed on the validation performance of our classification tasks, and only consider models that perform above that threshold. (Note that with this formulation, accuracy can be replaced with any domain specific notion of a high-quality model without modifying our approach.)

## 3.2 A Prior for Interpretable Models

Some model classes are generally amenable to human inspection (e.g. decision trees, rule lists, decision sets [9, 13]; unlike neural networks), but within those model classes, there likely still exist some models that are easier for humans to utilize than others (e.g. shorter decision trees rather than longer ones [23], or decision sets with fewer overlaps [13]). We want our model prior $p(M)$ to reflect this more nuanced view of interpretability.

We consider a prior of the form:

$$p(M) \propto \int_x \texttt{HIS}(x, M)p(x)dx \tag{2}$$

In our experiments, we will define $\texttt{HIS}(x, M)$ (human-interpretability-score) as:

$$\texttt{HIS}(x, M) = \begin{cases} 0, & \texttt{mean-RT}(x, M) > \texttt{max-RT} \\ \texttt{max-RT} - \texttt{mean-RT}(x, M), & \texttt{mean-RT}(x, M) \leq \texttt{max-RT} \end{cases} \tag{3}$$

where $\texttt{mean-RT}(x, M)$ (mean response time) measures how long it takes users to predict the label assigned to a data point $x$ by the model $M$, and $\texttt{max-RT}$ is a cap on response time that is set to a large enough value to catch all legitimate points and exclude outliers. The choice of measuring the time it takes to predict the model's label follows Doshi-Velez and Kim [8], which suggests this *simulation* proxy as a measure of interpretability when no downstream task has been defined yet; but any domain-specific task and metric could be substituted into our pipeline including error detection or cooperative decision-making.

### 3.3 A Prior for Arbitrary Models

In the interpretable model case, we can give a human subject a model $M$ and ask them questions about it; in the general case, models may be too complex for this approach to be feasible. In order to determine the interpretability of complex models like neural networks, we follow the approach in Ribeiro *et al.* [22], and construct a simple *local* model for each point $x$ by sampling perturbations of $x$ and training a simple model to mimic the predictions of $M$ in this local region. We denote this `local-proxy`$(M, x)$.

We change the prior in Equation 2 to reflect that we evaluate the `HIS` with the local proxy rather than the entire model:

$$p(M) \propto \int_x \texttt{HIS}(x, \texttt{local-proxy}(M, x))p(x)dx \tag{4}$$

We describe computational considerations for this more complex situation in Section 4.

## 4 Inference

Our goal is to find the MAP solution from Equation 1. Our overall approach will be to find a collection of models with high likelihood $p(X|M)$ and then perform model-based optimization [27] to identify which priors $p(M)$ to evaluate via user studies. Below, we describe each of the three main aspects of the inference: identifying models with large likelihoods $p(X|M)$, evaluating $p(M)$ via user studies, and using model-based optimization to determine which $p(M)$ to evaluate. The model from our set with the best $p(X|M)p(M)$ is our approximation to the MAP solution.

### 4.1 Identifying models with high likelihood $p(X|M)$

In the model-finding phase, our goal is to create a diverse set of models with large likelihoods $p(X|M)$ in the hopes that some will have large prior value $p(M)$ and thus allow us to identify the approximate MAP solution. For simpler model classes, such as decision trees, we find these solutions via running multiple restarts with different hyperparameter settings and rejecting those that do not meet our accuracy threshold. For neural networks, we jointly optimize a collection of predictive neural networks with different input gradient patterns (as a proxy for creating a diverse collection) [25].

### 4.2 Computing the prior $p(M)$

**Human-Interpretable Model Classes.** For any model $M$ and data point $x$, a user study is required for every evaluation of $\texttt{HIS}(x, M)$. Since it is infeasible to perform a user study for every value of $x$ for even a single model $M$, we approximate the integral in Equation 2 via a collection of samples:

$$
\begin{aligned}
p(M) &\propto \int_x \texttt{HIS}(x, M)p(x)dx \\
&\approx \frac{1}{N} \sum_{x_n \sim p(x)} \texttt{HIS}(x_n, M)
\end{aligned}
$$

In practice, we use the empirical distribution over the inputs $x$ as the prior $p(x)$.

**Arbitrary Model Classes.** If the model $M$ is not itself human-interpretable, we define $p(M)$ to be the integral over $\texttt{HIS}(x, \texttt{local-proxy}(M, x))$ where $\texttt{local-proxy}(M, x)$ locally approximates $M$ around $x$ (Equation 4). As before, evaluating $\texttt{HIS}(x, \texttt{local-proxy}(M, x))$ requires a user study; however, now we must determine a procedure for generating the local approximations $\texttt{local-proxy}(M, x)$.

We generate these local approximations via a procedure akin to Ribeiro *et al.* [22]: for any $x$, we sample a set of perturbations $x'$ around $x$, compute the outputs of model $M$ for each of those $x'$, and then fit a human-interpretable model (e.g. a decision-tree) to those data.

We note that these local models will only be nontrivial if the data point $x$ is in the vicinity of a decision boundary; if not, we will not succeed in fitting a local model. Let $B(M)$ denote the set of inputs $x$ that are near the decision boundary of $M$. Since we defined `HIS` to equal `max-RT` when

mean-RT$(x, M)$ is 0 as it does when no local model can be fit (see Equation 3), we can compute the integral in Equation 4 more intelligently by only seeking user input for samples near the model's decision boundary:

$$p(M) \propto \int_x \text{HIS}(x, \text{local-proxy}(M, x))p(x)dx \qquad (5)$$

$$= \left( \int_{x \in B(M)} p(x)dx \right) \cdot \left( \int_{x \in B(M)} \text{HIS}(x, \text{local-proxy}(M, x))\tilde{p}(x)dx \right)$$

$$+ \left( \int_{x \notin B(M)} p(x)dx \right) \cdot \text{max-RT}$$

$$\approx \left( \frac{1}{N'} \sum_{x_{n'} \sim p(x)} \mathbb{I}(x \in B(M)) \right) \cdot \left( \frac{1}{N} \sum_{x_n \sim \tilde{p}(x)} \text{HIS}(x_n, M) \right)$$

$$+ \left( \frac{1}{N'} \sum_{x_{n'} \sim p(x)} \mathbb{I}(x \notin B(M)) \right) \cdot \text{max-RT}$$

where $\tilde{p}(x) = p(x)/\int_{x \in B(M)} p(x)dx$. The first term (the volume of $p(x)$ in $B(M)$), and the third term (the volume of $p(x)$ not in $B(M)$) can be approximated without any user studies by attempting to fit local models for each point in $x$ (or a subsample of points). We detail how we fit local explanations and define the boundary in Appendix C.

### 4.3  Model-based Optimization of the MAP Objective

The first stage of our optimization procedure gives us a collection of models $\{M_1, ..., M_K\}$ with high likelihood $p(X|M)$. Our goal is to identify the model $M_k$ in this set that is the approximate MAP, that is, maximizes $p(X|M)p(M)$, with as few evaluations of $p(M)$ as possible.

Let $L$ be the set of all labeled models $M$, that is, the set of models for which we have evaluated $p(M)$. We estimate the values (and uncertainties) for the remaining unlabeled models—set $U$—via a Gaussian Process (GP) [21]. (See Appendix A for details about our model-similarity kernel.) Following Srinivas *et al.* [27], we use the GP upper confidence bound acquisition function to choose among unlabeled models $M \in U$ that are likely to have large $p(M)$ (this is equivalent to using the lower confidence bound to minimize response time):

$$a_{LCB}(M; L, \theta) \quad = \mu(M; L, \theta) - \kappa\sigma(M; L, \theta)$$
$$M_{\texttt{next}} \quad = \arg\min_{M \in U} a_{LCB}(M; L, \theta)$$

where $\kappa$ is a hyperparameter that can be tuned, $\theta$ are parameters of the GP, $\mu$ is the GP mean function, and $\sigma$ is the GP variance. (We find $\kappa = 1$ works well in practice.)

## 5  Experimental Setup

In this section, we provide details for applying our approach to four datasets. Our results are in Section 6.

**Datasets and Training Details**   We test our approach on a synthetic dataset as well as the mushroom, census income, and covertype datasets from the UCI database [7]. All features are preprocessed by z-scoring continuous features and one-hot encoding categorical features. We also balance the classes of the first three datasets by subsampling the more common class. (The sizes reported are after class balancing. We do not include a test set because we do not report held-out accuracy.)

- *Synthetic* ($N = 90,000$, $D = 6$, continuous). We build a data set with two noise dimensions, two dimensions that enable a lower-accuracy, interpretable explanation, and two dimensions that enable a higher-accuracy, less interpretable explanation. We use an 80%-20% train-validate split. (See Figure 1 in the Appendix.)

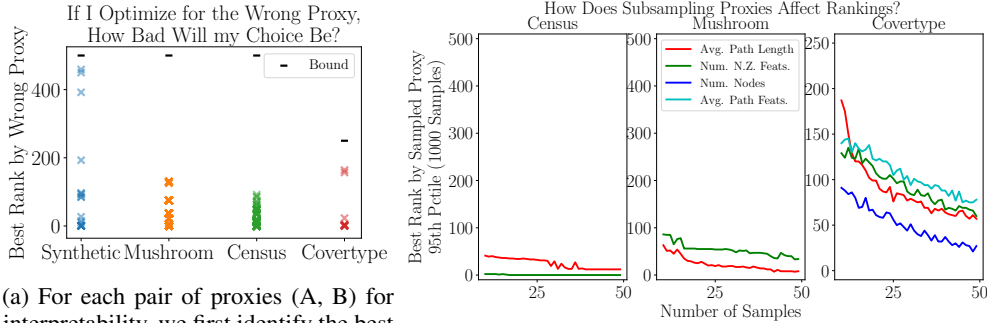

(a) For each pair of proxies (A, B) for interpretability, we first identify the best model if we only care about proxy A, then compute its rank if we now care about proxy B. This simulates the setting where we optimize for proxy B, but A is the true HIS. This value for each pair of proxies is plotted with an ×. The large ranking value indicates that sometimes proxies disagree on which models are good.

(b) Rank of the best model(s) by each proxy across multiple samples of data points ('N.Z.' denotes non-zero and 'feats.' denotes features). This simulates the setting where we compute HIS on a human accessible number of data points. The lines dropping below the high values in Figure 2a indicate that computing the right proxy on a human-accessible number of points is better than computing the wrong proxy accurately. This benefit occurs across all datasets and models, but it takes more samples for neural networks on Covertype than the others.

Figure 2: Determining interpretability on a few points is better than using the wrong proxy.

- *Mushroom* ($N = 8,000$, $D = 22$ categorical with 126 distinct values). The goal is to predict if the mushroom is edible or poisonous. We use an 80%-20% train-validate split.

- *Census* ($N = 20,000$, $D = 13$—6 continuous, 7 categorical with 83 distinct values). The goal is to predict if people make more than $50,000/year. We use their 60%-40% train-validate split.

- *Covertype* ($N = 580,000$, $D = 12$—10 continuous, 2 categorical with 44 distinct values). The goal is to predict tree cover type. We use a 75%-25% train-validate split.

Our experiments include two classes of models: decision trees and neural networks. We train decision trees for the simpler synthetic, mushroom and census datasets and neural networks for the more complex covertype dataset. Details of our model training procedure (that is, identifying models with high predictive accuracy) are in Appendix B. The covertype dataset, because it is modeled by a neural network, also needs a strategy for producing local explanations; we describe our parameter choices as well as provide a detailed sensitivity analysis to these choices in Appendix C.

**Proxies for Interpretability** An important question is whether currently used proxies for interpretability, such as sparsity or number of nodes in a path, correspond to some HIS. In the following we will use four different interpretability proxies to demonstrate the ability of our pipeline to identify models that are best under these different proxies, simulating the case where we have a ground truth measure of HIS. We show that (a) different proxies favor different models and (b) how these proxies correspond to the results of our user studies.

The interpretability proxies we will use are: mean path length, mean number of distinct features in a path, number of nodes, and number of nonzero features. The first two are local to a specific input $x$ while the last two are global model properties (although these will be properties of local proxy models for neural networks). These proxies include notions of tree depth [23] and sparsity [15, 20]. We compute the proxies based on a sample of $1,000$ points from the validation set (the same set of points is used across models).

**Human Experiments** In our human subjects experiments, we quantify $\text{HIS}(x, M)$ for a data point $x$ and a model $M$ as a function of the time it takes a user to simulate the label for $x$ with $M$. We extend this to the locally interpretable case by simulating the label according to $\text{local-proxy}(x, M)$. We refer to the model itself as the explanation in the globally interpretable case, and the local model as the explanation in the locally interpretable case. Our experiments are closely based on those in Narayanan *et al.* [19]. We provide users with a list of feature values for features used in the explanation and a graphical depiction of the explanation, and ask them to identify the correct prediction. Figure 3a in Appendix D depicts our interface. These experiments were reviewed and

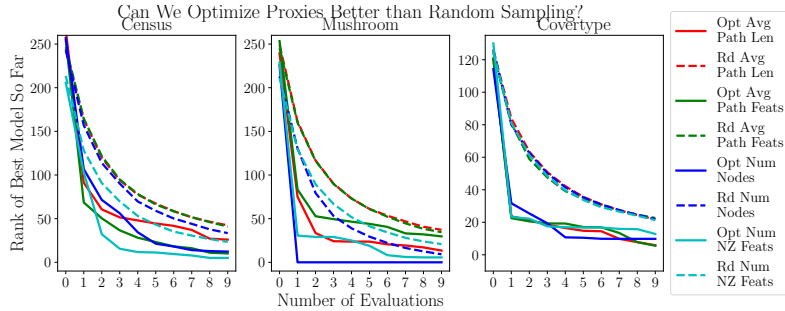

Figure 3: We ran random restarts of the pipeline with all datasets and proxies–denoted 'opt' (randomness from choice of start), and compared to randomly sampling the same number of models–denoted 'rd' (we account for models with the same score by computing the lowest rank of any model with that score). 'NZ' denotes non-zero and 'feats' denotes features. The fact that the solid lines stay below the corresponding dotted lines indicates that we do better than random guessing.

approved by our institution's IRB. Details of the experiments we conducted with machine learning researchers and details and results of a pilot study [not used in this paper] conducted using Amazon Turk are in Appendix D.

## 6 Experimental Results

**Optimizing different automatic proxies results in different models.** For each dataset, we run simulations to test what happens when the optimized measure of interpretability does not match the true HIS. We do this by computing the best model by one proxy–our simulated HIS, then identifying what *rank* it would have had among the collection of models if one of the other proxies–our optimized interpretability measure–had been used. A rank of 0 indicates that the model identified as the best by one proxy is the same as the best model for the second proxy; more generally a rank of $r$ indicates that the best model by one proxy is the $r$th-best model under the second proxy. Figure 2a shows that choosing the wrong proxy can seriously mis-rank the true best model. This suggests that it is not a good idea to optimize an arbitrary proxy for interpretability in the hopes that the resulting model will be interpretable according to the truly relevant measure. Figure 2a also shows that the synthetic dataset has a very different distribution of proxy mis-rankings than any of the real datasets in our experiments. This suggests that it is hard to design synthetic datasets that capture the relevant notions of interpretability since, by assumption, we do not know what these are.

**Computing the right proxy on a small sample of data points is better than computing the wrong proxy.** For each dataset, we run simulations to test what happens when we optimize the true HIS computed on only a small sample of points–the size limitation comes from limited human cognitive capacity. As in the previous experiment, we compute the best model by one proxy–our simulated HIS. We then identify what rank it would have had among the collection of models if the same proxy had been computed on a small sample of data points. Figure 2 shows that computing the right proxy on a small sample of data points can do better than computing the wrong proxy. This holds across datasets and models. This suggests that it may be better to find interpretable models by asking people to examine the interpretability of a small number of examples—which will result in noisy measurements of the true quantity of interest—rather than by accurately optimizing a proxy that does not capture the quantity of interest.

**Our model-based optimization approach can learn human-interpretable models that correspond to a variety of different proxies on globally and locally interpretable models.** We run our pipeline 100 times for 10 iterations with each proxy as the signal (the randomness comes from the choice of starting point), and compare to 1,000 random draws of 10 models. We account for multiple models with the same score by computing the lowest rank for any model with the same score as the model we sample. Figure 3 shows that across all three datasets, and across all four proxies, we do better than randomly sampling models to evaluate.

**Our pipeline finds models with lower response times and lower scores across all four proxies when we run it with human feedback.** We run our pipeline for 10 iterations on the census and

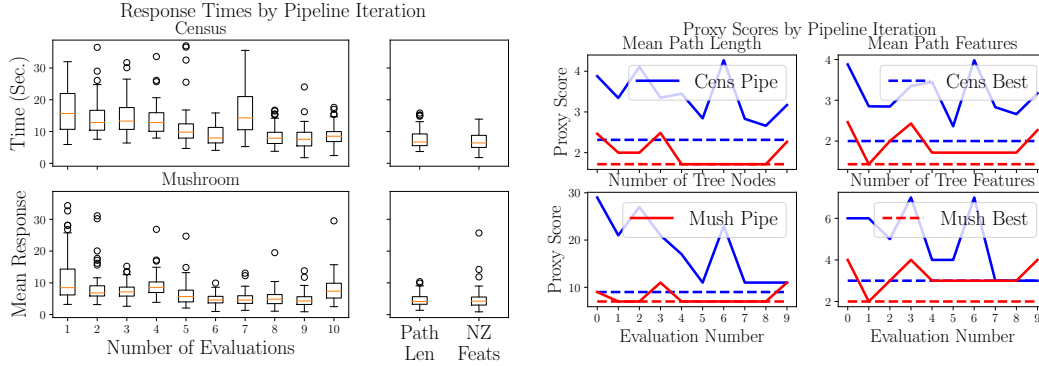

(a) We computed response times for each iteration of the pipeline on two datasets. Each data point is the mean response time for a single user. In both experiments, we see the mean response times decrease as we evaluate more models. We reach times comparable to those of the best proxy models. The last 2 models are our baselines ('NZ feats' denotes non-zero features).

(b) We computed the proxy scores for the model evaluated at each iteration of the pipeline. On the mushroom dataset, our approach converges to models with the fewest nodes and shortest paths, and on the census dataset, it converges to models with the fewest features. 'Mush' denotes the mushroom dataset and 'Cens' denotes the census dataset.

Figure 4: Human subjects pipeline results show a trend towards interpretability.

mushrooms datasets with human response time as the signal. We recruited a group of machine learning researchers who took all quizzes in a single run of the pipeline, with models iteratively chosen from our model-based optimization. Figure 4a shows the distributions of mean response times decreasing as we evaluate more models. (In Figure 3b in Appendix D we demonstrate that increases in speed from repeatedly doing the task are small compared to the differences we see in Figure 4a; these are real improvements in response time.)

**On different datasets, our pipeline converges to different proxies.** In the human subjects experiments above, we tracked the proxy scores of each model we evaluated. Figure 4b shows a decrease in proxy scores that corresponds to the decrease in response times in Figure 4a (our approach did *not* have access to these proxy scores). On the mushroom dataset, our approach converged to a model with the fewest nodes and the shortest paths, while on the census dataset, it converged to a model with the fewest features. This suggests that, for different datasets, different notions of interpretability are important to users.

## 7 Discussion and Conclusion

We presented an approach to efficiently optimize models for human-interpretability (alongside prediction) by directly including humans in the optimization loop. Our experiments showed that, across several datasets, several reasonable proxies for interpretability identify different models as the most interpretable; all proxies do not lead to the same solution. Our pipeline was able to efficiently identify the model that humans found most expedient for forward simulation. While the human-selected models often corresponded to some known proxy for interpretability, which proxy varied across datasets, suggesting the proxies may be a good starting point but are not the full story when it comes to finding human-interpretable models.

That said, the direct human-in-the-loop optimization has its challenges. In our initial pilot studies [not used in this paper] with Amazon Mechanical Turk (Appendix D), we found that the variance among subjects was simply too large to make the optimization cost-effective (especially with the between-subjects model that makes sense for Amazon Mechanical Turk). In contrast, our smaller but longer within-subjects studies had lower variance with a smaller number of subjects. This observation, and the importance of downstream tasks for defining interpretability suggest that interpretability studies should be conducted with the people who will use the models (who we can expect to be more familiar with the task and more patient).

The many exciting directions for future work include exploring ways to efficiently allocate the human computation to minimize the variance of our estimates $p(M)$ via intelligently choosing which inputs

$x$ to evaluate and structuring these long, sequential experiments to be more engaging; and further refining our model kernels to capture more nuanced notions of human-interpretability, particularly across model classes. Optimizing models to be human-interpretable will always require user studies, but with intelligent optimization approaches, we can reduce the number of studies required and thus cost-effectively identify human-interpretable models.

**Acknowledgments** IL acknowledges support from NIH 5T32LM012411-02. All authors acknowledge support from the Google Faculty Research Award and the Harvard Dean's Competitive Fund. All authors thank Emily Chen and Jeffrey He for their support with the experimental interface, and Weiwei Pan and the Harvard DTaK group for many helpful discussions and insights.

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
