[Supplementary Material]

# Supplementary Material for Human-in-the-Loop Interpretability Prior

**Isaac Lage**
Department of Computer Science
Harvard University
isaaclage@g.harvard.edu

**Andrew Slavin Ross**
Department of Computer Science
Harvard University
andrew_ross@g.harvard.edu

**Been Kim**
Google Brain
beenkim@google.com

**Samuel J. Gershman**
Department of Psychology
Harvard University
gershman@fas.harvard.edu

**Finale Doshi-Velez**
Department of Computer Science
Harvard University
finale@seas.harvard.edu

## A  Similarity Kernel for Models and GP parameters

Model-based optimization requires as input a notion of similarity. We use an RBF kernel between feature importances for decision trees, and between a gradient-based notion of feature importance for neural networks (average magnitude of the normalized input gradients for each class logit).

We use the scikit-learn implementation of Gaussian processes [1]. We set it to normalize $y$ automatically, restart the optimizer 10 times, and add $\alpha = 10^{-7}$ to the diagonal of the kernal at fitting to mitigate numerical issues. We used the default settings for all other hyperparameters, including the RBF kernel (on the model features above) for the covariance function.

## B  Experimental Details: Identifying a Collection of Predictive Models

We train decision trees for the synthetic, mushroom and census datasets with a test accuracy thresholds of $0.9$, $0.95$ and $0.8$ respectively. On the synthetic dataset, $0.9$ is slightly higher than the accuracy we can achieve on the interpretable dimensions. We make this choice to avoid learning the same, simple model over and over again. On the mushroom dataset, we can achieve a validate accuracy of 1 with decision trees, and on the Census dataset, we can achieve a validate accuracy of $0.83$ with decision trees. In both cases, we set the accuracy thresholds slightly below these numbers to ensure that we can generate distinct models that meet the accuracy threshold. For each of these, we train $500$ models.

To produce a variety of high-performing decision trees, we randomly sample the following hyperparameters: max depth [1-7], minimum number of samples at a leaf [1, 10, 100], max features used in a split [2 - num_features], and splitting strategy [best, random]. The first two hyperparameters are chosen to encourage simple solutions, while the last two hyperparameters are chosen to increase the diversity of discovered trees. We use the scikit-learn implementation [1], of decision trees and perform a post-processing step that removes leaf nodes iteratively when it does not decrease accuracy on the validation set (as in Wu *et al.* [4]).

We train neural networks for the covertype dataset with an accuracy threshold of $0.75$. We can achieve an accuracy of $0.71$ with logistic regression, so we set the threshold slightly above that to justify the use of more complex neural networks. For the neural network models, we randomly sample the following hyperparameters: L1 weight penalty [0, 0.0001, 0.001, 0.01], L2 weight penalty [0, 0.0001, 0.001, 0.01], L1 gradient regularization [0, 0.01], activation function [relu, tanh], architectures [three 100-node layers, two 100-node layers, one 100-node layer, one 25-node layer, one 250-node layer].

These are then jointly trained according to the procedure in Ross *et al.* [3] for 50 epochs for batch size 512 with Adam. (We train between 1 and 4 models simultaneously, another randomly sampled hyperparameter). For this dataset, we train 250 models.

Figure 1: We build a synthetic data set with two noise dimensions, two dimensions that enable a lower-accuracy, interpretable explanation, and two dimensions that enable higher-accuracy, less interpretable explanation. The purple data points are positive and the yellow are negative. Data points were generated for each set of two features independently, then points sharing the same label in all dimensions were randomly concatenated to form the final dataset.

## C    Experimental Details: Parameters and Sensitivity to Local Region Choices

We can ask humans to perform the simulation task directly using decision trees, but for the neural networks, we must train simple, local models as explanations (we use local decision trees). This procedure requires first sampling a local dataset for each point $x$ we explain. We modify the procedure in Ribeiro *et al.* [2] to sample $10,000$ points $x'$ in a radius around the point $x$ defined by its 20 nearest neighbors by Euclidean distance. We then binarize their predictions $M(x')$ to whether they match $M(x)$ and subsample the more common class to balance the labels. We do not fit explanations for points $x$ where the original sampled points $x'$ have a class imbalance greater than $0.75$; we consider these points not on the boundary. Finally, we return the simplest tree we train on this local dataset with accuracy above a threshold on a validate set. We randomly set aside $20\%$ of the sampled points for validation, and use the rest for training. (Note: if we were provided local regions by domain experts, we could use those.)

Our procedure for sampling points around some input $x$ uses two hyperparameters: a scaling factor for the empirical variance, and a mixing weight for the uniform distribution for categorical features that we use to adjust the empirical distribution of the point's 20 nearest neighbors. We use $0.01$ to weight the variance and $0.05$ to weight the categorical distributions. Finally, when training the trees, we set a local fidelity accuracy threshold of $90\%$ on a validation set and iteratively fit trees with larger maximum depth (up to depth 10) until one achieves this threshold. (We assume data points with local models deeper than this will not be interpretable, so fitting deeper trees will not improve our search for the most interpretable model.) We require at least 5 samples at each leaf. We use the scikit-learn implementation [1] to learn the trees and perform a post-processing step that removes leaf nodes iteratively when it does not decrease accuracy on the validation set (as in Wu *et al.* [4]).

How sensitive are the results to these choices? In Figure 2a, we first identify which of our $K = 250$ models would be preferred by each interpretability proxy if the local regions were determined by variance parameters set to $[0.001, 0.01, 0.1]$ and the mixing weights set to $[0.01, 0.05, 0.1]$ (9 combinations). Next, for each of those 9 models, we identify what *rank* it would have had among the $K$ models if one of the other variance or weight parameters had been used. Thus, a rank of $0$ indicates that the model identified as the best by one parameter setting is the same as the best model under the second setting; more generally a rank of $r$ indicates that the best model by one parameter setting is the $r$-best model under the second setting. The generally low ranks in the figure indicate agreement amongst the different choices for local parameter settings. The highest mismatch values for the number of nodes proxy all correspond to the variance scaling factor $0.1$ (which we do not use).

(a) We found the best model by each proxy for every setting of the region hyperparameters, and computed its rank by the same proxy for every other setting of the region hyperparameters. Each × corresponds to one of these pairs. The highest values all correspond to the variance scaling factor 0.1. The other two settings of this hyperparameter tend to agree on how to rank neural networks.

(b) We found the best model(s) by each proxy and computed their rank by the same proxy computed on a sample of data points. The comparable values of the lines across all three plots indicate that we need a similar number of samples to robustly rank neural networks for the smallest, middle and largest region settings (we do not include cross pairs).

Figure 2: Neural network local explanation sensitivity analysis

Do we need more points to estimate model rank correctly for any of these region settings? We find the best model(s) by each proxy, then we re-rank models using a small sample of points to compute the same proxy. We do this for the smallest, middle and largest settings of the local region parameters (we do not include cross-pairs of parameters in these results). Figure 2b shows that different hyperparameter settings require similar numbers of input samples $x$ to robustly approximate the integral for $p(M)$ in equation 4 for a variety of interpretability proxies substituted for HIS.

## D  Experimental Details: Human Subject Experiments

In our experiments, we needed to sample input points $x$ to approximate the prior $p(M)$ in equations 2 and 4. For globally interpretable models, we ask users about the same data points across all models to reduce variance. In the locally interpretable case, we would only conduct user studies for points near the boundary (in $B(M)$) and would thus sample points specific to each model's boundary. Each quiz contained 8 or 16 questions per model (8 for the pipeline experiments, 16 for the Amazon Turk experiments), with the order randomized across participants. There was also an initial set of 3 practice questions. If the participant answered these correctly, we allowed them to move directly to the quiz. If they did not, we gave them an additional set of 3 practice questions. We excluded people who answered fewer than 3 of each set of practice questions correctly from the Amazon Mechanical Turk experiments.

**Experiments with Machine Learning Graduate Students and Postdocs**  For the full pipeline experiment, models were chosen sequentially based on the subjects' responses. We collected responses from 7 subjects for each model in the experiment with the census dataset, and from 9 subjects for each model with the mushroom dataset.[1] Accuracies were all above $85\%$. We ran 10 iterations of the algorithm, each a quiz consisting of 8 questions about one model, and two evaluations at the end of the same format. We used the mean response time across users to determine $p(M)$. We did not exclude responses, and participants were compensated for their participation. Using the same set of subjects across all of these experiments substantially reduced response variance, although

(a) An example of our interface with a tree trained on the census dataset with the fewest non-zero features. In our experiments, we show people a decision tree explanation and a data point including only the features that appear in the tree. We then ask them to simulate the prediction according to the explanation.

(b) We asked a single user to take the same quiz 10 times to measure the effect of repetition on response time. The difference in mean response time between the first and last quiz is around 2 seconds. The $y$-axis scale is the same as that in 4a so the magnitude of the learning effect can be directly compared to the magnitude of the differences between models in our experiment.

Figure 3: Interface and learning effect

the smaller total number of subjects means we did not see statistically significant differences in our results.

**Experiments with Amazon Mechanical Turk**  We had initially hoped to use Amazon Mechanical Turk for our interpretability experiments. Here, we were forced to use a between-subjects design (unlike above), because it would be challenging to repeatedly contact previous participants to take additional quizzes as we chose models to evaluate based on the acquisition function.

In pilot studies, we collected 33 and 24 responses for the two models selected by the pipeline (the first had a medium mean path length, and the second had a high mean path length), after excluding people who did not get one of the two sets of practice questions right, or who took less than 5 seconds or more than 5 minutes for any of the questions on the quiz. The majority of respondents were between 18 and 34. We asked participants 16 questions with a 30 second break halfway through. We paid them $2 for completing the quiz.

The first model, which had a medium mean path length, had a mean time of 31.62s (28.61s - 34.63s), and a median time of 26.86s (23.09s - 30.63s) (standard error and median standard error in parentheses respectively). The second model with a high mean path length had a mean response time of 30.94s (28.66s - 33.22s), and a median response time of 30.32s (27.47s - 33.17s). These intervals are clearly overlapping. We could gather more samples to reduce the variance, but cost grows quickly; running one experiment with the end-to-end pipeline with these sample sizes would have cost around $1,000.

## Footnotes

[1]We recorded 2 extra responses for iteration number 4, and 2 fewer responses for iteration number 5 in the census experiment, and 1 extra response in iteration number 3 for the mushroom experiment due to a technical error discovered after the experiment, but we do not believe these affected our overall results. (Extra responses are from the same set of participants.)