[Reviews · NeurIPS 2018]

Reviewer 1



Overview: This paper introduces a bayesian framework for incorporating human feedback into the construction of interpretable models. The basic idea is to define a prior distribution over models that reflects the interpretability of a model M. In this paper, the (expected) time for a user to predict the label of points assigned by model M is used. Evaluating the prior is expensive; each call to p(M) requires a user study measuring HIS(x, M) for a sample of x values. To reduce expense, the authors propose to only query models that have a sufficiently high likelihood (high accuracy) given the observed data. The accompanying evaluation study shows that common proxies for interpretability (e.g., things like tree size) disagree among one another. Furthermore, which proxy is best aligned with the human-assessed interpretability prior differs across use cases (data sets). Comments: Overall I found this to be a very interesting paper that’s largely very well written. The idea of using user studies to elicit the relevant notion of “interpretability” is an interesting one. Furthermore, it opens the door to follow-up studies that are specialized to specific downstream tasks. (This sort of future direction could be mentioned in the discussion). I did get a bit lost in sections of the experimental results. I ask that the authors clarify the evaluation presented in Section 6. More specific comments are provided below. - In the introduction you write that interpretability depends on the “subjective experience of users.” More than that, it depends on the purpose for which an explanation is being desired. Assessing whether a model is fit-for-purpose would entail defining a specific task, which as you state is not something you do in this paper. Nevertheless I think it’s an important part of framing the general problem. - 3.3 prior for arbitrary models: I am not convinced by the idea that the interpretability of local proxies reflect the interpretability of the underlying arbitrary non-interpretable model. Have you considered framing this part of the analysis by saying there are two components: the base model (M) and the explanation model (e.g., local-proxy(M, x)). In this setup one could search over both explanation models and base models. Plausibly, an explanation model that works well for some M might work poorly for others. - 4.3, Appendix B: Similarity kernel. It seems like the similarity kernel is being defined in a way that two models that differ significantly in terms of their interpretability may nevertheless appear similar. Consider the setting where our model class is regression models. Correct me if I am wrong here, but I believe the proposed strategy for this class would amount to an RBF kernel between the estimated coefficient vectors. This would say that a sparse model with coefficient vector (w_1, …, w_k, 0, …, 0) is similar to (w_1, …, w_k, e_1, …, e_m) where the e_j are small coefficients. The same model could be deemed less similar to (w1, …, w_{k-1}, 0, w_k, 0, …, 0), depending on the choice of w_k. Yet from a structural and interpretability standpoint, the latter model, being sparse and just weighting a different coefficient, seems more similar than the first (which might have a large number of small non-zero entries). I suppose you can argue that if the e_j are all really small then they don’t really affect the model predictions much. But I’m still not sure that the similarity kernels are capturing quite the right type of similarity. - Section 6: I’m not sure I correctly understand lines 222 - 229. What is meant by “parameter setting”? Are these the parameters of your similarity kernel? Is the following interpretation correct: You have 500 models, each of which is being assessed by one of k proxies (proxy measures of interpretability, such as sparsity). To get an x for Figure 2(a) you look at the best model for each of the k proxies, and look at where it’s ranked by each of the remaining (applicable) k-1 proxies. In the worst case, the best model for proxy A could be the worst (rank 500) model for proxy B. Is this right? - Section 6: Computing risk proxy on small sample. Again, I’m not sure I correctly understand lines 233 - 241. How are you defining “the right proxy” here? Could you please further clarify the setup for this evaluation? - Appendix Figure 5: The y-axis scale is too large. No values exceed 20, but the axis goes up to 40. - There are a handful of typos throughout the paper. (E.g., line 254-255 “features non-zero features”) Unfortunately I didn’t note all of them down as I was reading. Please give the manuscript a couple of editing passes to weed these out.

Reviewer 2



The paper presents an approach to interpretability, where model optimization involves the estimation of an interpretability score by local proxies evaluated by humans. The integration of cognitive aspects with machine learning algorithms is an important aspect of the emerging field of interpretability. The objective is to include the interpretability requirement in the learning process in spite of the vagueness of the notion of interpretability. The present paper describes an interesting approach, which, as far as I know, is a new way of building cognitive aspects into a learning algorithm. Papers such as this one are important as they represent a response to the challenge raised by several recent position papers to address interpretability in a systematic manner. The difficulties described in the last section (like large variance in certain situations) give valuable information, as they point to phenomena which may be inherent in the human-in-the-loop approach in general and thus deserve to be better understood. Perhaps more of this discussion could be moved to the paper from the supplement.

Reviewer 3



== Update after author response == Thank you for the clarifications and answers. They all sound reasonable to me. I just have one comment, about this statement: "In this light, our work can be viewed as providing evidence for the use of these proxies in the future." I agree that your work provides evidence for these three proxies, but it seems like one could more directly (and probably more easily) test the proxies by giving humans different models that vary along those proxy scores and seeing how interpretable the models are. One would hope that human-in-the-loop optimization could give us a model, or tell us some principle about interpretability, that is harder to get from the proxy scores. At any rate, thanks again for the good paper. == Original review == The authors tackle the problem of learning interpretable models by directly querying humans about which models are more or less interpretable (as measured by the time taken by humans to perform forward simulation). In my opinion, this is a conceptually interesting and underexplored direction: research in interpretability often starts by declaring that some mathematical property equates to interpretability and then trying to calculate or optimize for that property without actually running any studies with actual humans. Studies like the present one therefore fill a much-needed gap in interpretability research. User studies take a lot of work. The strength of this paper is that it includes a broad variety of useful experimental investigations, e.g., comparing different proxies for interpretability, examining the feasibility of doing zeroth-order optimization with humans in the loop, and so on. The paper is well-written and it is clear that the authors have taken pains to be careful and scientific about their experiments. The main weakness of this paper, in my opinion, is that it lacks strong empirical results. In particular, from what I can tell, none of the human-optimized models actually beat models that are optimized by some easily-computable proxy for interpretability (e.g., mean path length in a decision tree). Indeed, the bulk of the paper talks about optimizing for interpretability proxies, though the title and rhetoric focus on the human-in-the-loop component. What, then, do we get out of human-optimization? The authors argue in line 249 that “Human response times suggest preferences between proxies that rank models differently.” This statement comes from Figure 3a, but the variance in the results shown in that figure seems so large that it’s hard to make any conclusive claims. Do you have some measure of statistical significance? Looking at Fig 4a, it seems like models that optimize for path length do as well as models that optimize for the number of non-zero features. Despite the above shortcomings, I think that this paper contains interesting ideas and is a solid contribution to human-in-the-loop ML research. It is hard to criticize the paper too much for not being able to show state-of-the-art results on such a challenging task, and this paper certainly stands out in terms of trying to involve humans in model building in a meaningful way. Other comments: 1) Is the likelihood formulation necessary? It complicates notation, and it’s not obvious to me that the SILF likelihood or a prior based on mean-response-time is particularly probabilistically meaningful. I think it might simplify the discourse to just say that we want accuracy to be above some threshold, and then we also want mean response time to be high, without dealing with all of the probabilistic notions. (e.g., as a reader, I didn’t understand the SILF measure or its motivation). 2) Does mean response time take into account correctness? A model that users can simulate quickly but incorrectly seems like it should be counted as uninterpretable. 3) Also, why is HIS proportional to the *negative* inverse of mean response time? 4) The local explanations part of the paper is the least convincing. It seems like a model that is locally explainable can be globally opaque; and none of the human results seem to have been run on the CoverType dataset anyway? Given that the paper is bursting at the seams in terms of content (and overflowing into the supplement :), perhaps leaving that part out might help the overall organization. 5) On local explanations, line 154 says: “We note that these local models will only be nontrivial if the data point x is in the vicinity of a decision boundary; if not, we will not succeed in fitting a local model.” Why is this true? Why can’t we measure differences in probabilities, e.g., “if feature 1 were higher, the model would be less confident in class y”?